

# Significant increase of summertime ozone at Mt. Tai in Central Eastern China: 2003-2015

Lei Sun[1], Likun Xue[1*], Tao Wang[2,1], Jian Gao[3], Aijun Ding[4], Owen R. Cooper[5,6], Pengju Xu[7], Zhe Wang[2], Xinfeng Wang[1], Liang Wen[1], Yanhong Zhu[1], Tianshu Chen[1], Lingxiao Yang[1,8], Yan Wang[8], Jianmin Chen[1,8], Wenxing Wang[1]

[1] Environment Research Institute, Shandong University, Ji'nan, Shandong, China

[2] Department of Civil and Environmental Engineering, Hong Kong Polytechnic University, Hong Kong, China

[3] Chinese Research Academy of Environmental Sciences, Beijing, China

[4] Institute for Climate and Global Change Research and School of Atmospheric Sciences, Nanjing University, Nanjing, Jiangsu, China

[5] Cooperative Institute for Research in Environmental Sciences, University of Colorado, Boulder, Colorado, United States

[6] NOAA Earth System Research Laboratory, Boulder, Colorado, United States

[7] School of Geography and Environment, Shandong Normal University, Ji'nan, Shandong, China

[8] School of Environmental Science and Engineering, Shandong University, Ji'nan, Shandong, China

*To whom correspondence should be addressed: Likun Xue: xuelikun@sdu.edu.cn

**Abstract.**

Tropospheric ozone ($O_3$) is a trace gas playing key roles in atmospheric chemistry, air quality and climate change. In contrast to North America and Europe, China has limited long-term records of surface $O_3$ that can be used to establish trends. In this study, we compiled the available observations of $O_3$ at Mt. Tai – the highest mountain over the North China Plain (NCP), and analyzed their seasonal and diurnal behavior as well as the trends over 2003–2015. The summertime climatological air mass



transport pattern was established by back trajectory calculation and a subsequent cluster analysis. A significant increase of surface $O_3$ ($p < 0.01$) in the summertime from 2003 to 2015 was derived from a linear regression analysis, with increasing rates of 1.7 ppbv $yr^{-1}$ for June and 2.1 ppbv $yr^{-1}$ for the July–August period. Analysis of satellite trace gas retrievals indicates that the $O_3$ increase was mainly

due to the increased emissions of $O_3$ precursors, in particular volatile organic compounds (VOCs). An important finding is that the emissions of nitrogen oxides ($NO_x$) have diminished since 2011, but the increase of VOCs appears to have enhanced the ozone production efficiency and contributed to the observed $O_3$ increase in northern China. This study provides direct evidence that controlling $NO_x$ alone, in the absence of VOC controls, is not sufficient to reduce regional $O_3$ levels in North China. In addition,

the ozone observations at this regionally representative mountain site are ideal for evaluating global and regional scale chemical transport models.

## 1. Introduction

Ozone ($O_3$) in the troposphere is a trace gas of great importance for climate and air quality. It is the principal precursor of the hydroxyl radical (OH) which plays a central role in atmospheric chemistry

(Seinfeld and Pandis, 2006), and the third most important greenhouse gas contributing to the warming of the Earth (IPCC, 2013). At ground level, high levels of $O_3$ have adverse effects on human health and ecosystem productivity (National Research Council, 1991; Monks et al., 2015). In the troposphere, the ambient $O_3$ burden is the product of the flux from the stratosphere (Stohl et al., 2003), dry deposition, and net photochemical production involving the reactions of nitrogen oxides ($NO_x$) with carbon

monoxide (CO) and volatile organic compounds (VOCs) in the presence of sunlight (Crutzen, 1973; Ma et al., 2002). Increases in anthropogenic emissions of ozone precursors have driven changes in the tropospheric $O_3$ abundances, both globally and regionally (Cooper et al., 2014; Monks et al., 2015; The Royal Society, 2008). Conversely, the changing tropospheric $O_3$ may also pose significant feedbacks to the environment and climate (e.g., Shindell et al, 2012; Stevenson et al, 2013). Therefore, the long-term

changes (or trends) of tropospheric $O_3$ has long been a topic of great interest in the atmospheric sciences.



Since the 1970s, long-term measurements of surface $O_3$ (and $O_3$ precursors) have been increasingly carried out worldwide, mostly in North America and Europe (Cooper et al., 2014; and references therein). The existing knowledge of tropospheric $O_3$ trends has been recently reviewed (Cooper et al., 2014; Monks et al., 2015; UNEP, 2011). Overall, upward trends have been recorded around the world

since the 1970s, but trends over the past two decades have varied regionally. In Europe, the surface $O_3$ in rural or remote areas, usually regarded as the regional background $O_3$, rose until the year 2000 but has since leveled off or decreased (Oltmans et al., 2013; Parrish et al., 2012). In the eastern US, summertime $O_3$ at most rural and urban stations has decreased over 1990–2010 (Cooper et al., 2014; Lefohn et al., 2010). In the western US, extreme ozone events have decreased in urban areas but rural

ozone has either remained steady or even increased during springtime (Cooper et al., 2012; Oltmans et al., 2013; Parrish et al., 2012). In contrast, the limited measurements in East Asia have shown significant increasing trends in the regional background $O_3$ since the 1990s (Cooper et al., 2014; and references therein).

In comparison with North America and Europe, investigations of long-term $O_3$ trends are scarce in

China, where rapid urbanization and industrialization has occurred over the past three decades. Although significant increasing trends of surface $O_3$ have been derived from long-term air quality monitoring or series of short-term measurements at urban and suburban sites in Beijing (Tang et al., 2009; Zhang et al., 2014), Hong Kong (Xue et al., 2014), and Taiwan (Lin et al., 2010), long-term observations covering more than ten years in rural or remote areas (indicative of the regional

background $O_3$ trend) remain very limited (Cooper et al., 2014). Wang et al. (2009) reported the first long-term continuous observations of Chinese surface $O_3$ at a regional background site in southern China (Hok Tsui), and indicated an average increase of 0.58 ppbv $yr^{-1}$ during 1994–2007. Xu et al. (2015) recently reported another continuous ozone record (1994–2013) at Mt. Waliguan, a Global Atmospheric Watch station in western China, and also found significant positive trends with 0.15–0.27

25     ppbv $yr^{-1}$ for daytime $O_3$ and 0.13–0.29 ppbv $yr^{-1}$ for nighttime $O_3$. Based on the MOZAIC commercial aircraft measurements, Ding et al. (2008) derived an ozone increase of ~2% per year from 1995 to 2005 for the lower troposphere above Beijing by analyzing the difference between observations from



1995–2000 and 2000–2005. Despite the valuable information obtained from the abovementioned efforts, additional studies are required to improve our understanding of tropospheric $O_3$ trends across the rapidly developing China.

In recent years, China has phased in a series of stringent air quality control measures. Following the successful reductions in sulfur dioxide ($SO_2$) emissions in 2006 (Lu et al., 2010), China has recently launched a national programme to reduce $NO_x$ emissions during its "Twelfth Five-Year Plan" (2011–2015) (China State Council, 2011). However, to our knowledge, few controls have been placed on VOC emissions in China. Rather, anthropogenic VOC emissions have continued to increase (Bo et al., 2008; Wang et al., 2014). Therefore, it is of great interest and critical importance to evaluate the effect of the current control policy (i.e., controlling $NO_x$ with little action for VOCs) on regional $O_3$ and other secondary air pollution problems in China.

Mt. Tai (36.25 °N, 117.10 °E; 1534 m above sea level (a.s.l.)) is the highest mountain in the center of the North China Plain (NCP; see Fig. 1) – a fast developing region facing severe air pollution. During daytime, its summit is well within the planetary boundary layer (PBL), and the site is therefore regionally representative of the NCP region (Kanaya et al., 2013). Since 2003, several field measurement campaigns have been conducted at the site, with surface $O_3$ of major interest. In this paper, we analyze all available ozone and ozone precursor observations at Mt. Tai to understand the summertime $O_3$ characteristics, including trends. In the following sections, we first present the seasonal and diurnal ozone variations followed by the climatological air mass transport pattern in summer; we then derive the $O_3$ trend (or systematic change) during 2003–2015 using linear regression; we finally elucidate the key factors affecting the $O_3$ trends by examining satellite and in-situ observed trace gas data. Our analysis demonstrates a significant increase of summertime surface ozone at this important regional site in northern China, and indicates the urgent need of VOC control in China to reduce regional ozone pollution.

## 2. Observational data set

The observations analyzed in the present study are summarized in Table 1. It comprises several sets



of field observations from different periods. The earliest ozone measurements at Mt. Tai were made from July–November 2003 (Gao et al., 2005), and the longest observations lasted for three years from June 2006 to June 2009 (despite a data gap in January–February 2007 owing to instrument maintenance). In recent years, we conducted intensive measurements at Mt. Tai during summer, i.e.

June–August, of 2014 and 2015. In addition, another set of 3-year measurements was carried out from March 2004 to May 2007, as an international joint effort between Japanese and Chinese scientists (Kanaya et al., 2013); monthly average $O_3$ data are taken from this work. As most of the measurements are available for the period of June–August each year, we focus on the summertime $O_3$ in this study.

Two study sites have been used for field observations at Mt. Tai. One was the Mt. Tai

Meteorological Observatory (Site 1) at the summit with an altitude of 1534 *m* a.s.l., and the other was in a hotel (Site 2) that is ~1 *km* to the northwest of Site 1 and slightly lower (1465 *m* a.s.l.; see *Fig. S1* for the site locations). Such elevations position these sites either within the PBL in the afternoon in summer, or in the free troposphere during the night. Although Mt. Tai is a famous tourism spot, both sites are located in the less frequently visited zones. Hence the impact of local anthropogenic emission should be

small, and the data collected are believed to be regionally representative. Details of these sites have been described elsewhere (Site 1: Gao et al., 2005; Kanaya et al., 2013; Site 2: Guo et al., 2012; Shen et al., 2012). For our data set, most of the measurements were taken at Site 1, and only the intensive campaign in 2014 took place at Site 2.

All measurements were implemented using standard techniques, which were detailed in the

previous publications (e.g., Gao et al., 2005; Xue et al., 2011). Briefly, $O_3$ was measured using a commercial ultraviolet photometric instrument (*Thermo Environment Instruments (TEI), Model 49C*) with a detection limit of 2 ppbv and a precision of 2 ppbv. CO was monitored with a gas filter correlation, non-dispersive infrared analyzer (*Teledyne Advanced Pollution Instrumentation, Model 300E*), with automatic zeroing every two hours. This technique has a detection limit of 30 ppbv and a

precision of 1% for a level of 500 ppbv. NO and $NO_2$* were measured by a chemiluminescence analyzer equipped with an internal MoO catalytic converter (*TEI, Model 42C*), with a detection limit of 0.4 ppbv and precision of 0.4 ppbv. Inter-comparison with a highly selective photolytic $NO_2$ detection





approach indicated that the $NO_2$* measured with MoO conversion significantly overestimated $NO_2$ (i.e., up to 130% in afternoon hours), and $NO_2$* actually represented a major fraction (60%–80%) of $NO_y$ at Mt. Tai (Xu et al., 2013). During the measurements, the $O_3$ analyzer was calibrated routinely (i.e., quarterly for the 3-year observations in 2006–2009, and before and after the other campaigns) by an

ozone primary standard (*TEI, Model 49PS*). For CO and $NO_x$*, zero and span calibrations were performed weekly during 2006–2009 and every three days during the intensive campaigns. Meteorological data including temperature, relative humidity (RH), and wind vectors were obtained from the Mt. Tai Meteorological Observatory, where Site 1 is located.

It is noteworthy that some portion of this long-term data set has been reported previously. Gao et al.

(2005) analyzed the measurements of $O_3$ and CO from July–November 2003 and examined their diurnal variations and relations to backward trajectories. Kanaya et al. (2013) reported the observations in 2004–2007 and examined the processes influencing the seasonal variations and regional pollution episode. The major objective of the present study is to compile all of the available $O_3$-related observations at Mt. Tai and to establish the trend (or systematic change), if any, of ambient $O_3$ levels in

the past decade at this unique site, regionally representative of the NCP.

## 3.   Results and discussion

### 3.1.   Seasonal and diurnal variations from more recent data

Figure 2 depicts the seasonal variation of surface $O_3$ at Mt. Tai derived from the more recent year-round observations from 2006–2009. Overall, $O_3$ shows higher levels in the warm season, i.e.

April–October, compared to the cold season, i.e. November–March, with two peaks in June and October. In addition to the high temperatures and intense solar radiation (especially in June), biomass burning is believed to be another factor shaping the $O_3$ maximums in June and October, both of which are major harvest seasons of wheat and corn in northern China. The significant impacts of biomass burning on air quality at Mt. Tai have been evaluated by Yamaji et al. (2010) and Suthawaree et al. (2010). Modeling

and data analysis studies usually averaged the $O_3$ levels in June, July and August (JJA) to represent the overall summertime condition (e.g., Lin et al., 2008 and 2009). For the Mt. Tai case, however, it is



noticeable that the $O_3$ concentrations in July and August were substantially lower than those in June. This is attributable to the more humid weather and greater precipitation in July and August in this region (see Table 2 for the RH condition). Inspection of meteorological conditions day by day also indicated a frequency of cloudy days (i.e., with RH ≥ 95%) of ~25% in June and of ~51% during July–August. In

the following analyses, therefore, we assess the ozone characteristics separately for June and July–August.

Shown in the upper panel of Fig. 2 is the frequency of the maximum daily 8-hour average $O_3$ mixing ratios (MDA8 $O_3$) exceeding the national ambient air quality standard (i.e., 75 ppbv; Class II). Although located in a relatively remote mountain-top area, the observed $O_3$ pollution at Mt. Tai was

rather serious in the warm season, with frequencies of the $O_3$-exceedence days of over 45% throughout April–October. In particular, the occurrence of $O_3$-exceedence days was as high as 89% in June. These results demonstrate the severe $O_3$ pollution situation across the North China Plain.

Figure 3 illustrates the well-defined diurnal variations of surface $O_3$ with a trough in the early morning and a broad peak lasting from afternoon to early evening, which is commonly observed at

polluted rural sites. As the summit of Mt. Tai is well above the planetary boundary layer at nighttime, the $O_3$ concentrations during the latter part of the night (e.g., 2:00–5:00 LT) are usually considered to represent the regional background $O_3$. Comparing the diurnal profiles in June and July–August clearly reveals the significantly higher regional background $O_3$ in June (with a mean difference of ~17 ppbv). On the other hand, the daytime $O_3$ build-up, defined as the increase in $O_3$ concentrations from the

early-morning minimum to the late-afternoon maximum, may reflect the potential of regional $O_3$ formation. For the Mt. Tai case, the average daytime $O_3$ build-up was 22 ppbv in June and 15 ppbv in July–August, indicating the stronger photochemical ozone production in June. Hence, the more intense photochemistry and higher regional background bring about the more serious $O_3$ pollution in June at Mt. Tai.

Another remarkable feature of surface $O_3$ at Mt. Tai is the relatively high nighttime levels, with average concentrations of 75–85 ppbv in June and 60–70 ppbv during July–August. This should be the





composite result of the residual $O_3$ produced in the preceding afternoon in the boundary layer, less $O_3$ loss from NO titration, and long-range transport of processed regional plumes. The latter was evidenced by the coincident evening $NO_2^*$ maximums and low NO levels, as shown in Fig. 4. Inspection of the time series day by day also reveals the frequent transport of photochemically aged air masses containing

elevated concentrations of $O_3$ (over 100 ppbv), CO and $NO_2^*$ to the study site during the late evening (figures not shown). Similarly, the MOZAIC aircraft measurements have also found ~60 ppbv on average of $O_3$ at around 1500 $m$ a.s.l. over Beijing at 5:00–6:00 LT in summer (i.e., May–July; Ding et al., 2008), which is comparable to what we observed at Mt. Tai. These results imply the existence of the $O_3$-laden air in the nocturnal residual layer over the NCP region. Moreover, Ding et al. (2008) also

showed in their Figure 11 that the $O_3$ enhancement extended from the surface up to about 2 km, further evidence that during the daytime Mt. Tai should be within the boundary layer and is sampling at a vertical level where ozone enhancements are expected.

### 3.2. Impact of long-range transport

Long-range transport associated with synoptic weather and large-scale circulations is an important

factor for the variation of ozone in rural areas (Wang et al., 2009; Ding et al., 2013; Zhang et al., 2016). To elucidate the history of air masses sampled at Mt. Tai, we analyzed the summertime climatological air mass transport pattern during 2003–2015 with the aid of cluster analysis of back trajectories. The detailed methodology has been documented by Wang et al. (2009) and Xue et al. (2011). Three-dimensional 72-hour back trajectories were computed four times a day (i.e., 2:00, 8:00, 14:00 and

20:00 LT) for June–August with the Hybrid Single-Particle Lagrangian Integrated Trajectory model (HYSPLIT, v4.9; Draxler et al. 2009), with an endpoint of 300 $m$ above ground level exactly over Mt. Tai. All the trajectories were then categorized into a small number of major groups with the HYSPLIT built-in cluster analysis approach.

A total of five air mass types were extracted for the summer period, with four identified for June

and July–August respectively. These air mass types are named according to the regions they traversed, and are described as follows: "Marine and East China" (M&EC) – air masses from the east passing over





the ocean and polluted central eastern China; "Northeast China" (NEC) – air masses from the north passing over Northeast China; "Central China" (CC) – air masses from the south moving slowly over central China; "Southeast China" (SEC) – air masses from the south moving fast from southeast China; "Mongolia and North China" (M&NC) – air masses from the northwest passing over Mongolia and central northern China.

The above identified major types of air masses are presented in Fig. 5. In June, M&EC was most frequent (57%), followed by CC (26%), NEC (9%) and M&NC (8%; only identified in June). During the July–August period, the most frequent air mass type was still M&EC (36%), then CC (29%) and NEC (29%), with a minor fraction of SEC (6%; only identified in July–August). Overall, the transport patterns in June and July–August are quite similar, and it is evident that southerly and easterly pathways dominated the air mass transport to Mt. Tai in summer. Such patterns are believed to be driven by the summer Asian monsoon (Ding et al., 2008).

The chemical signatures of the different air masses were also inspected and summarized in Table 3. The air masses of M&EC, CC and NEC, which passed over several polluted regions of eastern China, contained higher abundances of $O_3$ (with averages of 89–94 ppbv in June and 64–77 ppbv in July–August), CO and $NO_2*$. In comparison, the more aged air masses of M&NC and SEC showed relatively lower concentrations of $O_3$ ($78\pm21$ ppbv for M&NC in June and $58\pm17$ ppbv for SEC in July–August) and its precursors (except for $NO_2*$ in the SEC air mass). Therefore, the regions with the greatest influence on $O_3$ at Mt. Tai in summer are primarily located in the southern and eastern parts of central eastern China.

### 3.3. Ozone Trend

Figure 6 presents the monthly average hourly $O_3$ and MDA8 $O_3$ mixing ratios in June and July–August whenever available from 2003–2015 at Mt. Tai. The least square linear regression analysis reveals the significant increase of surface $O_3$ at Mt. Tai since 2003. Monthly mean $O_3$ values based on hourly data increased at rates of $1.7\pm1.0$ ppbv $yr^{-1}$ ($\pm95\%$ confidence intervals) in June and $2.1\pm0.9$ ppbv $yr^{-1}$ in July–August, and the increases were statistically significant ($p < 0.01$). For the monthly





means based on MDA8 $O_3$, the fewer available data points (as we only have monthly average data during 2004–2005 from Kanaya et al., 2013) likely reduced the significance of the trend in June, with a positive by statistically insignificant increase (rate = 1.4±1.9 ppbv yr$^{-1}$; $p = 0.12$). However the site had a significant positive trend in July–August (rate = 2.2±1.2 ppbv yr$^{-1}$; $p < 0.01$). Therefore we conclude

5     that summertime surface $O_3$ levels at Mt. Tai have increased over the period 2003–2015.

Given the fact that Mt. Tai is above the PBL at night when there is no photochemistry, the ambient $O_3$ levels before dawn (e.g., 2:00–5:00 LT) are representative of the regional background $O_3$. The diurnal variation in Fig. 3 shows a slight but steady decrease in $O_3$ concentrations overnight, which should arise from dry deposition. It was assumed that dry deposition was essentially the same every

10    year and did not affect the derived trends. Figure 7 shows the monthly averaged late-night $O_3$ mixing ratios in June and July–August available from 2003–2015 at Mt. Tai. Again, positive trends were found. The rate of increase was quantified at 1.9±1.8 ppbv yr$^{-1}$ ($p = 0.04$, significant) in June and 1.1±1.2 ppbv yr$^{-1}$ ($p = 0.06$, insignificant) during July–August. The increase of regional background likely explains the observed $O_3$ rise at Mt. Tai in June and also accounts for the majority of the increase during

July–August. These results indicate the significant increase of surface $O_3$ in summer on the regional scale across northern China.

Table 4 compares the surface and lower tropospheric ozone trends available in East Asia in recent decades. Two aspects are particularly noteworthy from this comparison. First, most studies have deduced significant positive trends demonstrating the broad increase of tropospheric $O_3$ over East Asia,

especially in China. This pattern is distinct from that found in Europe and the eastern U.S., where $O_3$ levels have begun to decrease or level off since the 1990s or 2000s (e.g., Cooper et al., 2014; Lefohn et al., 2010; Oltmans et al., 2013; Parrish et al., 2012). The ozone increase in East Asia is expected due to the rapid economic growth and increasing anthropogenic emissions of $O_3$ precursors in the past three decades (*e.g.,* Ohara et al., 2007). Second, the magnitude of $O_3$ increase is quite heterogeneous in

different regions and the fastest rise was found in the NCP. For example, the rates of $O_3$ increase were in the range of 0.54-0.58 ppbv yr$^{-1}$ in Hong Kong (1994-2007, Wang et al., 2009; 2002-2013, Xue et al., 2014), 0.54 ppbv yr$^{-1}$ in Taiwan (1994-2007, Lin et al., 2010), 0.26-0.55 ppbv yr$^{-1}$ over South Korea





(1990-2010, Lee et al., 2013; 1999-2010, Seo et al., 2014), 0.22-0.37 ppbv yr$^{-1}$ in major Japanese metropolitan areas (1990-2010; Akimoto et al., 2015), 0.64 ppbv yr$^{-1}$ at Mt. Happo, Japan (1991-2011, summer scenario; Parrish et al., 2014), and 0.15 ppbv yr$^{-1}$ at Mt. Waliguan, a GAW station in western China (1994-2013, summer scenario; Xu et al., 2015). According to the MOZAIC aircraft observations,

in comparison, Ding et al. (2008) have reported the PBL O$_3$ increases of ~1 ppbv yr$^{-1}$ for the annual average and ~3 ppbv yr$^{-1}$ for the summer afternoon peaks over the period of 1995-2005. Zhang et al. (2014) analyzed their field measurements at an urban site in Beijing during 2005-2011 and quantified an increasing rate of 2.6 ppbv yr$^{-1}$ for the daytime average O$_3$ in summer. Comparable rates of O$_3$ increase (1.7-2.1 ppbv yr$^{-1}$) were determined in the present study from the measurements of longer time coverage

and at a more regionally representative mountain site, affirming the significant rise of surface O$_3$ levels over the NCP region. Furthermore, the magnitude of O$_3$ increase in the NCP region is also among the highest records currently reported in the world (Cooper et al., 2012 and 2014; Lin et al., 2014; Parrish et al., 2014).

### 3.4. Roles of meteorology and anthropogenic emissions

Meteorology and changes in anthropogenic emissions may affect tropospheric O$_3$ directly (Monks et al., 2015; Mao and Talbot, 2004). To further elucidate the factors which have driven the increase of surface O$_3$ at Mt. Tai, we examined the long-term variations in both meteorological condition and anthropogenic emissions of O$_3$ precursors during the past decade. Table 2 summarizes the summertime mean meteorological conditions including average temperature, RH, and prevailing wind direction

recorded at Mt. Tai over the period of 2003–2015. Though some variability is clearly shown, no systematic change was found with regard to the overall meteorological conditions. We also explored the air mass transport pattern deduced from cluster analysis of back trajectories (see Section 3.2) year by year over 2003–2015 (*Table S1*). Despite the quite large year-to-year variability, again, no systematic change in the air mass transport pattern was indicated during the target period. Although the impact of

meteorology on tropospheric ozone is very complex and might not be quantified by such a simple analysis, the significant increase of surface O$_3$ observed at Mt. Tai should not primarily arise from the change in meteorological conditions.



We then analyzed the satellite retrievals of formaldehyde (HCHO) and $NO_2$ to track the variations

in the abundances of $O_3$ precursors (i.e., $NO_x$ and VOCs) during the study period. HCHO was adopted

as an indicator of VOCs since it is a major oxidation product of VOCs and because of the availability of

satellite products. The satellite data were obtained from SCIAMACHY for 2003–2011 and GOME-2(B)

from 2013 onwards, with the Level-2 products taken from the TEMIS archive (Tropospheric Emission

Monitoring    Internet    Service;    http://www.temis.nl/index.php).    Considering    the    geographical

representativeness of Mt. Tai, a larger domain (32 °–38 ° N, 115 °–120 ° E; see *Fig. 5*) was selected to

process the regional mean satellite data. The monthly averaged HCHO and $NO_2$ column densities in

June and July–August from 2003–2015 are documented in Fig. 8. Significant positive trends are seen

for HCHO, with rates of 2.7% $\pm$ 2.2% ($p = 0.02$) for June and 2.2% $\pm$ 1.4% ($p < 0.01$) for July–August,

indicative of the strong increase of VOCs in this region. This result agrees with the emission inventory

estimates which showed significant increases of anthropogenic VOC emissions in China in the past

decades (Bo et al., 2008; Wang et al., 2014), and is consistent with the lack of nationwide VOC

controls.

What is more interesting is the two-phase variation of the $NO_2$ column, showing a significant

increase first from 2003 to 2011 (June: 4.8% $\pm$ 3.4%, $p = 0.01$; July–August: 7.7% $\pm$ 3.6%, $p < 0.01$)

and a decrease afterwards (see Fig. 8b). These satellite observations agree very well with the bottom-up

emission inventory estimates, which clearly showed a break point occurring in 2011 in the

anthropogenic $NO_x$ emissions of China (see *Fig. S2*). China has just launched a national $NO_x$ control

programme during its "Twelfth Five-Year Plan" (i.e., 2011–2015) (China State Council, 2011). The

strict control measures are very efficient and have resulted in an immediate reduction of $NO_x$ emissions,

as affirmed by both emission inventories and satellite retrievals. Furthermore, the reduced levels of $NO_x$

in the most recent five years were also evidenced by our limited in-situ $NO_x$* measurements at Mt. Tai.

As shown in Fig. 9, the ambient $NO_2$* levels in 2014 and 2015 were indeed substantially lower than

those measured in the previous years before 2010.

From the above analyses, the $O_3$ increase between 2003 and 2011 is easy to understand in light of



the consistent increase of both $NO_x$ and VOCs. For the later period, i.e. after 2011, in comparison, opposite trends have taken place with $NO_x$ decreasing but VOCs still increasing. The observed continuing $O_3$ rise suggests that the reduction of $NO_x$ is not adequate to reduce the ambient $O_3$ levels, with a background of increasing VOCs. We then evaluated the ozone production efficiency (OPE) for

the air masses sampled at Mt. Tai. OPE is usually derived from the regression slope of the scatter plots of $O_3$ versus $NO_z$ (Trainer et al., 1993), and is a useful metric to infer how efficient $O_3$ is produced per oxidation of unit of $NO_x$ (e.g., Wang et al., 2010; Xue et al., 2011). As $NO_y$ (and thus $NO_z$) is not routinely measured in the present study, $NO_2^*$ is used instead of $NO_z$ to infer the OPE values. It should be reasonable considering that our measured $NO_2^*$ significantly overestimated true $NO_2$ and actually

contained a large fraction of $NO_z$, especially in the afternoon period when $NO_z$ was at its maximum with $NO_2$ at the minimum (Xu et al., 2013). The scatter plots of $O_3$ versus $NO_2^*$ during the afternoon hours (i.e., 12:00–18:00 LT) available from 2006–2015 at Mt. Tai are presented in Fig. 10. The OPE values in 2014–2015 (i.e., 9.6-15.0) were significantly higher than those determined during 2006–2009 (i.e., 3.6-7.9). This demonstrates the greater ozone production efficiency in recent years with increasing

VOCs. These results indicate that although $NO_x$ in China has been reduced since 2011, little action on VOC control has led to increased emissions of VOCs, which could make $O_3$ formation more efficient per unit of $NO_x$. As a consequence, ambient $O_3$ levels have been rising in northern China. We conclude that control of VOCs is urgently needed, in addition to the ongoing strict $NO_x$ control, to mitigate regional $O_3$ pollution in China.

**4.   Summary**

In the present study, we compiled all field observations of $O_3$ and $O_3$ precursors ever collected at Mt. Tai, the highest point of the NCP, and found a significant increase of surface $O_3$ in summertime over 2003–2015 at this regionally representative site. The rising $O_3$ levels are primarily attributable to the increasing anthropogenic emissions of $O_3$ precursors, especially VOCs. Since 2011, $NO_x$ has been

reduced efficiently by a strict national control programme, as evidenced by emission inventory estimates, satellite remote sensing and field observations. At the same time, VOC emissions increased, which enhanced the $O_3$ production efficiency and resulted in an overall $O_3$ increase in northern China.



This study provides direct evidence that the current Chinese control programme, focusing on $NO_x$ alone with little action for VOCs, is not sufficient for mitigating regional $O_3$ pollution, and calls on the implementation of VOC controls as soon as possible. Similar to the U.S., China phased in its air pollution control measures resulting in decreasing $SO_2$ emission after 2006 and $NO_x$ emission after 2011.

It is foreseen that the national VOC control will be launched very soon. Thus, follow-up long-term measurements are required to evaluate the response of ambient $O_3$ to the upcoming VOC control and to provide observational constraints to evaluate global and regional chemical transport models.

### Acknowledgements

The authors thank Steven Poon and Wei Nie for their contributions to the field study, and the staff

of the Mt. Tai Meteorological Observatory for the logistics and help during the field measurements. We are also grateful to the NOAA Air Resources Laboratory for providing the HYSPLIT model and meteorological data, and the European Space Agency for the free distribution of SCIAMACHY and GOME-2(B) satellite data through the TEMIS website (http://www.temis.nl/index.php). This work was supported by the National Natural Science Foundation of China (project no.: 41275123) and the Qilu

Youth Talent Programme of Shandong University.

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



**Table 1.** The observations at Mt. Tai analyzed in the present study.

| Year | Month | Observed species | Data source |
|---|---|---|---|
| 2003 | Jul.-Nov. | $O_3$, CO | Gao et al. (2005) and our study |
| 2004 | Jun.-Aug. | $O_3$ | Kanaya et al.( 2013)[a] |
| 2005 | Jun.-Aug. | $O_3$ | Kanaya et al.( 2013)[a] |
| 2006 | Jun.-Dec. | $O_3$, $NO_2$*, NO | Our study |
| 2007 | Mar.-Dec. | $O_3$, NO, $NO_2$*, CO | Our study [b] |
| 2008 | Jan.-Dec. | $O_3$, NO, $NO_2$*, CO | Our study |
| 2009 | Jan.-Jun. | $O_3$, NO, $NO_2$*, CO | Our study |
| 2014 | Jun.-Aug. | $O_3$, $NO_x$*, CO | Our study [c] |
| 2015 | Jun.-Aug. | $O_3$, NO, $NO_2$*, CO | Our study [c] |

[a] Monthly average data were taken from Kanaya et al. (2013).

[b] Valid NO and $NO_2$* data were only available in March-April and July-December, and CO was available in March-April in 2007.

5      [c] Note that the intensive measurement periods were 6 June – 3 July & 24 July – 26 August in 2014 and 14 June – 8 August in 2015.



**Table 2.** Summary of meteorological conditions recorded at Mt. Tai in June and July-August over 2003-2015[a].

| Year | June | | | July-August | | |
|---|---|---|---|---|---|---|
| | Temperature (°C) | RH (%) | Prevailing WD | Temperature (°C) | RH (%) | Prevailing WD |
| 2003 | 15.4±3.1 | 70.9±19.2 | SW | 17.3±2.9 | 88.3±16.3 | SW |
| 2004 | 15.4±3.9 | 74.4±21.2 | SW | 17.0±2.5 | 86.8±15.5 | SSW |
| 2005 | 18.1±3.2 | 65.0±22.0 | SSW | 17.4±2.9 | 86.8±16.3 | SW |
| 2006 | 17.2±2.9 | 67.8±21.4 | SSW | 18.3±2.0 | 88.3±16.3 | SSW |
| 2007 | 17.0±2.8 | 71.4±27.3 | E | 17.8±2.3 | 85.2±21.1 | E |
| 2008 | 15.0±3.6 | 76.9±20.0 | S | 17.0±2.3 | 86.3±14.6 | S |
| 2009 | 17.9±3.1 | 57.8±22.2 | SSW | 17.4±2.9 | 78.7±21.8 | S |
| 2010 | 15.9±3.6 | 80.1±18.7 | SW | 18.2±2.7 | 91.0±16.9 | SW |
| 2011 | 16.6±2.7 | 72.6±21.1 | SSW | 17.6±2.5 | 87.6±16.1 | S |
| 2012 | 16.8±3.2 | 71.0±23.8 | SW | 18.2±2.4 | 89.5±17.9 | E |
| 2013 | 16.1±3.1 | 79.8±19.6 | SW | 19.3±2.4 | 88.2±15.6 | SW |
| 2014 | 15.3±2.9 | 79.9±17.6 | SW | 16.5±2.3 | 86.7±14.6 | SW |
| 2015 | 15.8±2.8 | 70.5±21.0 | SW | 18.0±2.1 | 86.1±16.9 | SW |

[a] Average and standard deviations are provided.

**Table 3.** Statistics of $O_3$, $NO_2$* and CO in different air mass categories [a]

 [a] The ppbv;

| June | | | | July-August | | | | | | unit is |
|---|---|---|---|---|---|---|---|
| Air mass [b] | $O_3$ | $NO_2$* | CO | Air mass [b] | $O_3$ | $NO_2$* | CO |
| M&EC | 92±27 | 7.2±5.3 | 500±300 | M&EC | 64±22 | 3.6±2.9 | 370±180 |
| CC | 89±24 | 6.3±5.0 | 550±300 | CC | 77±21 | 3.3±2.5 | 440±180 |
| NEC | 94±25 | 7.0±4.5 | 380±180 | NEC | 73±22 | 4.5±4.2 | 380±180 |
| M&NC | 78±21 | 4.0±3.3 | 280±180 | SEC | 58±17 | 4.1±2.4 | 350±80 |

average and standard deviations are provided.

[b] Refer to *Figure* 5 and Section 3.2 for the derivation and description of the air mass types.



**Table 4.** Summary of surface and lower tropospheric ozone trends recorded in East Asia.

| Station | Site type | Period | Rate of change (ppbv yr$^{-1}$) | Reference |
|---|---|---|---|---|
| Mt. Tai | rural | 2003-2015 (summer) | 1.7 ± 1.0 (June) | This study |
| | | | 2.1 ± 0.9 (July-August) | |
| Beijing | rural (MOZAIC) | 1995-2005 | ~1 (annual average) | Ding et al. (2008) |
| Hong Kong (Hok Tsui) | rural | 1994-2007 | ~3 (summer afternoon) | Wang et al. (2009) |
| | | | 0.58 | |
| Hong Kong | urban & suburban | 2002-2013 (autumn) | 0.54 ± 0.49 | Xue et al. (2014) |
| Lin'an | rural | 1991-2006 | 2.7% (summer daily maximum) | Xu et al. (2008) |
| Waliguan | remote | 1994-2013 (summer) | 0.15 ± 0.19 | Xu et al. (2015) |
| Taiwan (Yangming) | rural | 1994-2007 | 0.54 ± 0.21 | Lin et al. (2010) |
| Mt. Happo, Japan | rural | 1991-2011 (summer) | 0.64 ± 0.40 | Parrish et al. (2012) |
| Tokyo, Japan | urban & suburban | 1990-2010 | 0.31 ± 0.02 | Akimoto et al. (2015) |
| Nagoya, Japan | urban & suburban | 1990-2010 | 0.22 ± 0.05 | Akimoto et al. (2015) |
| Osaka/Kyoto, Japan | urban & suburban | 1990-2010 | 0.37 ± 0.03 | Akimoto et al. (2015) |
| Fukuoka, Japan | urban & suburban | 1990-2010 | 0.37 ± 0.04 | Akimoto et al. (2015) |
| South Korea | 124 urban sites average | 1999-2010 | 0.26 | Seo et al. (2014) |
| South Korea | 56 urban sites average | 1990-2010 | 0.48 ± 0.07 (annual average) | Lee et al. (2013) |
| | | | 0.55 ± 0.13 (summer) | |





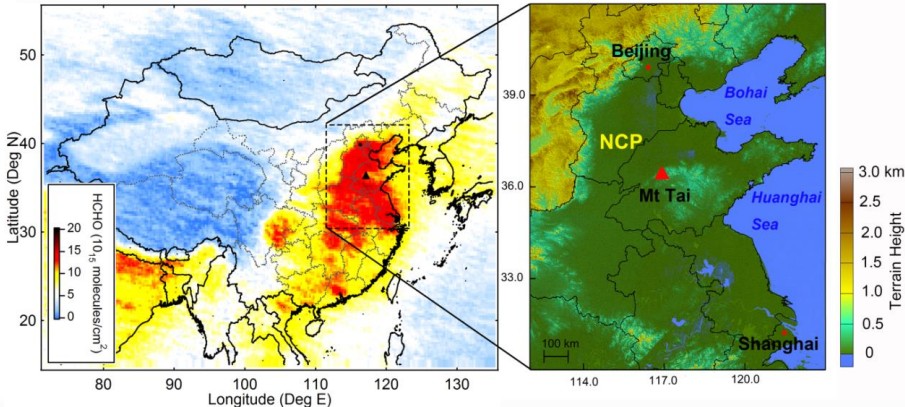

**Figure 1.** Geographical map showing the North China Plain and the location of Mt. Tai. The left map is color-coded by the HCHO column density in the summer period (JJA; 2003–2014) retrieved from SCIAMACHY and GOME-2(B).

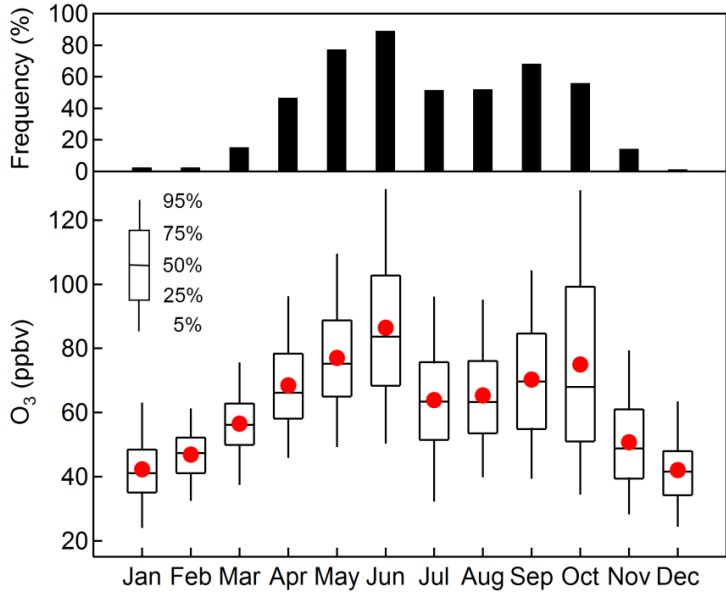

**Figure 2.** Seasonal variation of surface $O_3$ mixing ratios at Mt. Tai derived from the continuous observations from 2006–2009. Red dots indicate the monthly average $O_3$ concentrations. Shown in the upper panel is the frequency of the MDA8 $O_3$ exceeding the Chinese national ambient air quality standard, i.e. 75 ppbv (Class II).



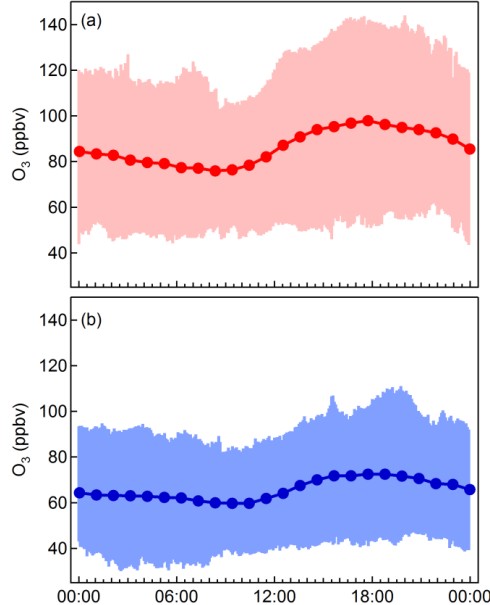

**Figure 3.** Average diurnal variations of surface $O_3$ at Mt. Tai in (a) June and (b) July-August derived from the continuous observations from 2006–2009. The shaded area indicates the 5th and 95th percentiles of the data.

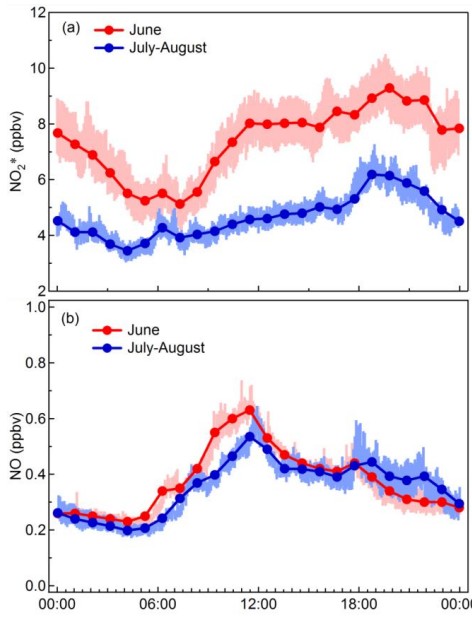

**Figure 4.** Average diurnal variations of (a) $NO_2^*$ and (b) NO at Mt. Tai in June and July-August derived from the continuous observations from 2006–2009. The shaded area indicates the standard error of the mean.



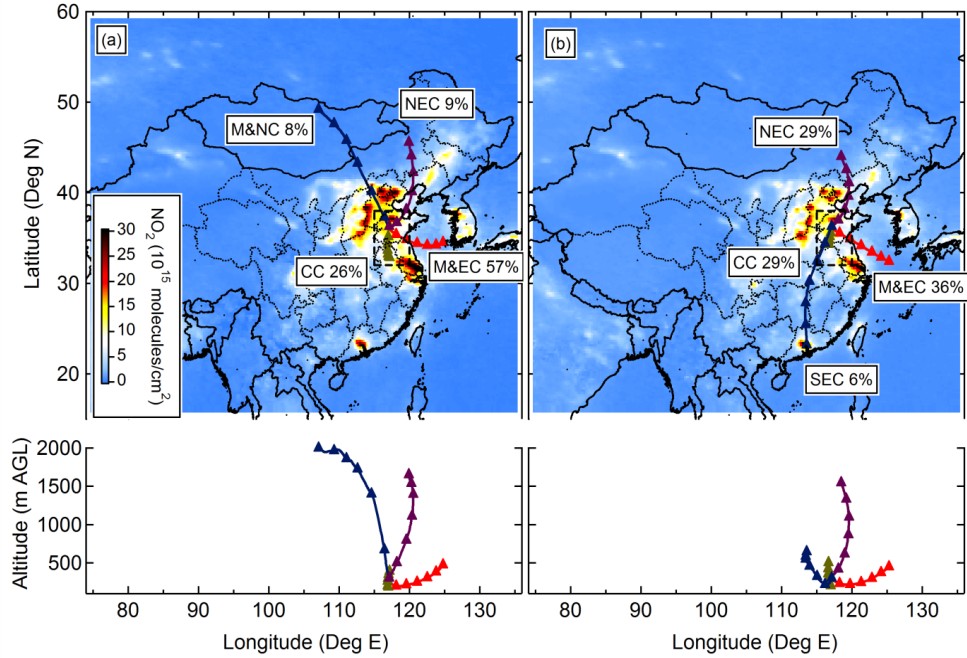

**Figure 5.** Climatological air mass transport pattern at Mt. Tai in (a) June and (b) July-August over 2003-2015. The maps are color-coded by the $NO_2$ column density retrieved from SCIAMACHY (2003-2011) and GOME-2 (B) (2013-2015). The box (dashed line) refers to the domain for which the satellite retrievals were averaged. Five major air masses: (1) M&EC: Marine and East China, (2) NEC: Northeast China, (3) CC: Central China, (4) SEC: Southeast China, (5) M&NC: Mongolia and North China.





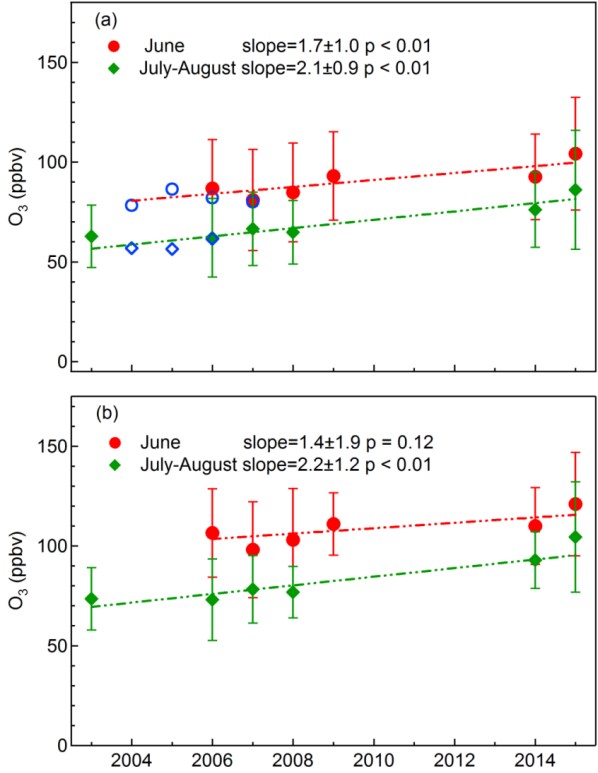

**Figure 6.** Monthly averaged (a) 1-hour and (b) MDA8 $O_3$ mixing ratios at Mt. Tai in June and July-August over 2003-2015. Error bars indicate the standard deviation of the mean. The blue open circles and squares represent the data taken from Kanaya et al. (2013). The fitted lines are derived from the least square linear regression analysis with the slopes ($\pm$ 95% confidence intervals) and $p$ values annotated.

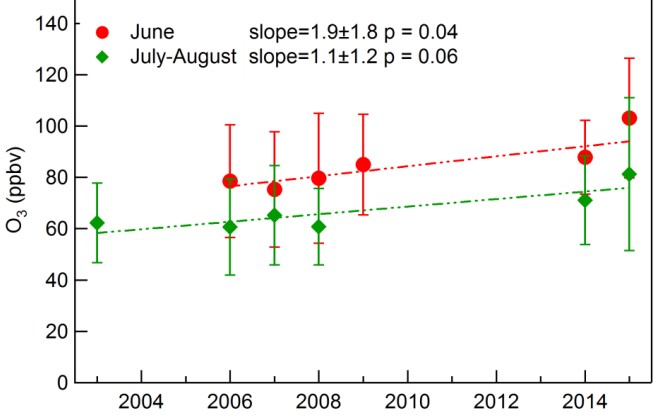

**Figure 7.** Monthly averaged nighttime $O_3$ mixing ratios (inferring the regional background $O_3$) at Mt. Tai in June and July-August over 2003-2015. Error bars indicate the standard deviation of the mean. The fitted lines are derived from the least square linear regression analysis with the slopes ($\pm$ 95% confidence intervals) and $p$ values annotated.



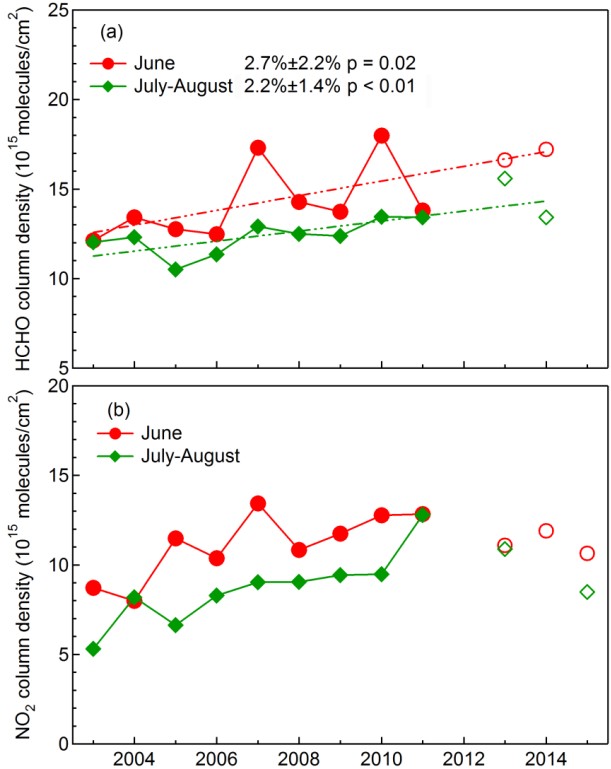

**Figure 8.** Monthly average column density of (a) formaldehyde and (b) NO₂ retrieved from SCIAMACHY (2003–2011; solid markers) and GOME-2 (B) (2013–2015; open markers) for the target domain (32 °–38 °N, 115 °–120 °E). For formaldehyde, the fitted lines are derived from the least square linear regression analysis with the slopes (±95% confidence intervals) and $p$ values also shown.





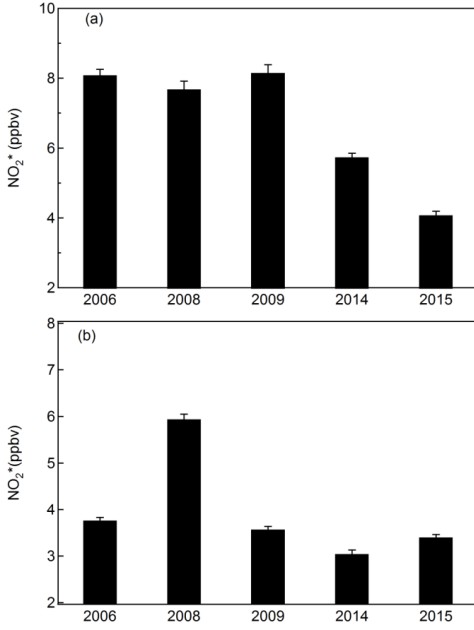

**Figure 9.** Monthly averaged $NO_2*$ concentrations measured at Mt. Tai in (a) June and (b) July-August during 2006–2015. Error bars indicate the standard error of the mean. Note that the data point in June 2014 is for $NOx^*$ instead of $NO_2*$.

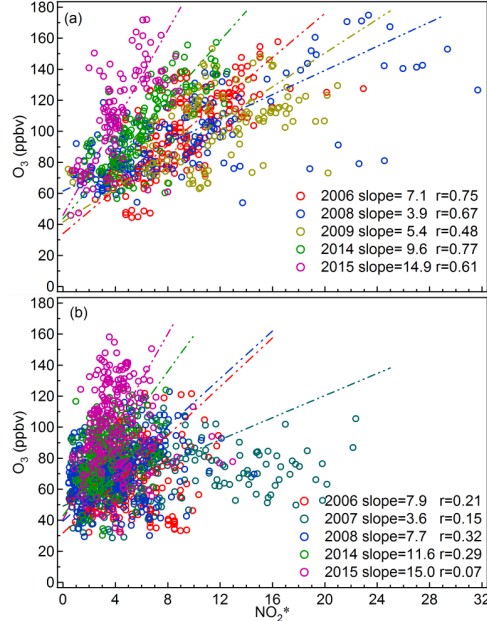

**Figure 10.** Scatter plots of $O_3$ versus $NO_2*$ at Mt. Tai in (a) June and (b) July–August during 2006–2015. Only the afternoon data (i.e., 12:00-18:00 local time) were used for this analysis. Note that the $NO_2*$ data in June 2014 stand for $NOx^*$ ($NO_x*=NO+NO_2*$), of which NO usually presents a minor fraction. The slopes are determined by the reduced major axis (RMA) method.