# Peer review of "Significant increase of summertime ozone at Mount Tai in Central Eastern China"

_Atmospheric Chemistry and Physics, 2016_

## Referee Comment (RC1) · Anonymous Referee #1 · 4 May 2016

General

Sun et al. present a long-term analysis of ozone at Mt. Tai mountain station over the North China Plain in China during 2003-2015. This study focuses on the variation and trend of summertime ozone, and points out that the increased VOC emissions other than changes in meteorology or NOx emissions are responsible for the elevated summertime ozone at the monitoring site. The manuscript is well structured, the methodology is appropriate and properly conducted, and the conclusion drawn is fully supported by the data presented. It is recommended that this manuscript be accepted after consideration of the few minor comments that follow.

Minor comments

1. P2L10, I don't see any relevance of this sentence to the study. And why the observations at Mt. Tai are 'ideal' for evaluating CTMs?

2. P2L23, 'the changing tropospheric O3' is confusing. Suggest using 'the changes in tropospheric O3'.

3. P4L16, suggest replacing 'at the site' with 'at this site'.

4. P4L19-20, suggest replacing ';' with '.' and capitalizing the following word.

5. P8L2, what does 'the latter' refer to, 'less O3 loss' or 'long-range transport of processed regional plumes'? Also, NO2* peaks around 20:00 p.m. but NO reaches the lowest level around 6:00 a.m., isn't it?

6. P9L10, according to the frequency, no air mass transported from the south in June and more is from the north than the south (29

7. P9L19, again, why does the southern part of central eastern China greatly impact ozone at Mt. Tai?

8. Have you looked at the contribution of stratosphere to troposphere transport to the surface ozone at Mt. Tai?

9. Note a in Table 3 needs to be reformatted.

---

## Referee Comment (RC2) · Anonymous Referee #2 · 1 Jun 2016

The paper describes the trends in ozone levels in North China. As pointed out by the authors there are very few studies of ozone trends in the rapidly developing region of China. The paper uses both new and previously reported data to look at ozone trends over ten years. There are several issues that need to be addressed/answered

1. The seasonal variation shows a typical spring maximum in ozone. This is a e result of maximum stratospheric-tropospheric exchange and whilst photochemical of ozone is maximised in the summer other loss process are enhanced and thus a typical dip in ozone is observed. There is no mention of start-trop exchange and the impact on ozone and this needs to be addressed.

2. More details are required on how the air mass types for the impact of long range transport, i.e. what is the statistical significance of the 5, why not, for example 7, a few

more details on the cluster analysis need to be provided

3. The authors suggest that because the site is above the PBL O3 levels before dawn (e.g., 2:00–5:00 LT) are representative of the regional background O3. This could be the case, however all the that can be said is that the site is dynamically disconnected from the PBL, how can they be sure that it is representative of the background?

4. There is work by Clapp and Jenkin Atmospheric Environment 35 (2001) 6391–6405 that has shown that it is possible to quantify the regional contribution to oxidant (i.e. NOx independent) from plots of Ox vs. NOx. This work also looked at the contribution as a function of season too. This analysis should also be carried out for data presented in this present study and compare to the data presented in figure 7.

5. The authors use HCHO as a marker for anthropogenic VOCs. There are not only anthropogenic sources of HCHO, there are also secondary in situ production routes, such as from CH4. Given that there are not only anthropogenic sources of HCO, who will this impact on their analysis?

6. they are using satellite retrievals to infer PBL HCHO. Satellite retrials in the PBL are notoriously difficult. What a priori assumptions have been made for their retrievals? More importantly what are the errors and what impact will it have on there results?

---

## Author Comment (AC1)

**Response to Reviewer 1**

Sun et al. present a long-term analysis of ozone at Mt. Tai mountain station over the North China Plain in China during 2003-2015. This study focuses on the variation and trend of summertime ozone, and points out that the increased VOC emissions other than changes in meteorology or NOx emissions are responsible for the elevated summertime ozone at the monitoring site. The manuscript is well structured, the methodology is appropriate and properly conducted, and the conclusion drawn is fully supported by the data presented. It is recommended that this manuscript be accepted after consideration of the few minor comments that follow.

Response: we appreciate the reviewer for the positive comments and helpful suggestions. We have revised the manuscript accordingly and here address individually the review comments. For clarity, the reviewer's comments are listed below in black italics, while our responses and changes in manuscript are shown in blue and red, respectively.

 P2L10, I don't see any relevance of this sentence to the study. And why the observations at Mt. Tai are 'ideal' for evaluating CTMs?

Response: this sentence has been removed from the revised manuscript.

2. P2L23, 'the changing tropospheric  $O_3$ ' is confusing. Suggest using 'the changes in tropospheric  $O_3$ '.

Response: changed.

3. P4L16, suggest replacing 'at the site' with 'at this site'.

Response: changed.

4. P4L19-20, suggest replacing ';' with '.' and capitalizing the following word.

Response: changed.

5. P8L2, what does 'the latter' refer to, 'less  $O_3$  loss' or 'long-range transport of processed regional plumes'? Also,  $NO_2^*$  peaks around 20:00 p.m. but NO reaches the lowest level around 6:00 a.m., isn't it?

Response: 'the latter' refers to the long-range transport of processed regional plumes. The

higher  $NO_2^*$  levels suggest the transport of anthropogenic pollution to the mountain site (note that the  $NO_2^*$  includes not only  $NO_2$  but also some higher oxidized  $NO_Z$  species), and the relatively low NO concentrations (it doesn't matter if NO was at its lowest) indicate that the air masses had been chemically processed (or aged). In the revised manuscript, the original statement has been revised as follows to clarify this issue.

"The transport of reginal plume was evidenced by the coincident evening  $NO_2^*$  (including  $NO_2$  and some higher oxidized nitrogen compounds) maximums and relatively low NO levels (indicative of the aged air mass), as shown in Fig. 4."

6. P9L10, according to the frequency, no air mass transported from the south in June and more is from the north than the south.

Response: from the back trajectories, both "M&EC" and "CC" air masses originated from the southeast and south. These two types of air masses accounted for 83% of the total in June. Thus it could be said that the air mass transport was dominated by the southerly and easterly air flows. We have elaborated this by the following statements in the revised manuscript.

"Marine and East China" (M&EC) – air masses from the southeast passing over the ocean and polluted central eastern China;

"Overall, the transport patterns in June and July–August are quite similar, and it is evident that southerly and easterly air flows (e.g., M&EC and CC) dominated the air mass transport to Mt. Tai in summer."

7. P9L19, again, why does the southern part of central eastern China greatly impact ozone at *Mt. Tai?*

Response: this argument is supported by the analyses of frequency and chemical composition of air mass types. Specifically, the "M&EC", "CC" and "NEC" air masses occurred the most frequently. These air masses had passed over the southern and eastern parts of central eastern China prior to arriving at Mt. Tai, and contained relatively higher concentrations of  $O_3$  and  $O_3$  precursors. These results indicated that the southern and eastern parts of central eastern China significantly affect the ozone pollution at Mt. Tai. The original statement has been revised as follows.

"In view of the higher frequency and higher  $O_3$  levels of the M&EC, CC and NEC air masses, it could be concluded that the regions with the greatest influence on  $O_3$  at Mt. Tai in summer are primarily located in the southern and eastern parts of central eastern China."

8. Have you looked at the contribution of stratosphere to troposphere transport to the surface ozone at Mt. Tai?

Response: the stratosphere to troposphere exchange (STE) generally occurs at its maximum in spring. In summer, the  $O_3$  pollution levels at Mt. Tai should be primarily affected by the photochemical processes of anthropogenic pollution in the planetary boundary layer (Li et al., 2008). As the present study mainly focused on the summertime  $O_3$  trend, we didn't consider the contribution of the STE process.

Li, J., Wang, Z., Akimoto, H., Yamaji, K., Takigawa, M., Pochanart, P., Liu, Y., Tanimoto, H., and Kanaya, Y. Near-ground ozone source attributions and outflow in central eastern China during MTX2006. *Atmos. Chem. Phys.* **8**, 7335-7351, 2008.

9. Note a in Table 3 needs to be reformatted.

Response: done.

---

## Author Comment (AC2)

**Response to Reviewer 2**

The paper describes the trends in ozone levels in North China. As pointed out by the authors there are very few studies of ozone trends in the rapidly developing region of China. The paper uses both new and previously reported data to look at ozone trends over ten years. There are several issues that need to be addressed/answered.

Response: we thank the reviewer for the helpful comments. Below we address the specific comments and will revise the manuscript accordingly. For clarity, the reviewer's comments are listed below in black italics, whilst our responses and changes in manuscript are shown in blue and red respectively.

1. The seasonal variation shows a typical spring maximum in ozone. This is a result of maximum stratospheric-tropospheric exchange and whilst photochemical of ozone is maximised in the summer other loss process are enhanced and thus a typical dip in ozone is observed. There is no mention of start-trop exchange and the impact on ozone and this needs to be addressed.

Response: in the revised manuscript, we have addressed the impact of stratosphere-troposphere exchange on the springtime  $O_3$  levels. The discussion of  $O_3$  seasonal variation has been revised as follows.

"Overall,  $O_3$  shows higher levels in the warm season, i.e. April–October, compared to the cold season, i.e. November–March, with two peaks in June and October. The elevated  $O_3$  levels in April and May should be affected by the stratosphere-troposphere exchange process which usually occurs at its maximum in the spring season (Yamaji et al., 2006). In addition to the high temperatures and intense solar radiation (especially in June), biomass burning is believed to be another factor shaping the  $O_3$  maximums in June and October, both of which are major harvest seasons of wheat and corn in northern China. The significant impacts of biomass burning on air quality over the North China Plain during June have been evaluated by a number of studies (Lin et al., 2009; Yamaji et al., 2010; Suthawaree et al., 2010). It is also noticeable that the  $O_3$ concentrations in July and August at Mt. Tai are substantially lower than those in June. This is attributed in part to the more humid weather and greater precipitation in July and August in this region (see Table 2 for the RH condition)."

2. More details are required on how the air mass types for the impact of long range transport, i.e.

what is the statistical significance of the 5, why not, for example 7, a few more details on the cluster analysis need to be provided.

Response: the cluster analysis was performed with the HYSPLIT built-in cluster analysis module. Specifically, total spatial variance (TSV) and the variance between each component trajectory (SPVAR, Spatial Variance) are calculated to choose the optimum number of clusters (Draxler et al., 2009). The most appropriate solution could be identified just before the large increase in TSV according to the relationship between TSV and the number of clusters. We have added some details in the revised manuscript, and the revised descriptions are as follows.

"To elucidate the history of air masses sampled at Mt. Tai, we analyzed the summertime climatological air mass transport pattern during 2003-2015 with the aid of cluster analysis of back trajectories. The NCEP reanalysis GDAS data and archive data (http://ready.arl.noaa.gov/archives.php) were used to compute trajectories during 2003-2004 and 2005-2015. The detailed methodology has been documented by Wang et al. (2009) and Xue et al. (2011). Briefly, three-dimensional 72-hour back trajectories were computed four times a day (i.e., 2:00, 8:00, 14:00 and 20:00 LT) for June-August with the Hybrid Single-Particle Lagrangian Integrated Trajectory model (HYSPLIT, v4.9; Draxler et al. 2009), with an endpoint of 300 m above ground level exactly over Mt. Tai. All the trajectories were then categorized into a small number of major groups with the HYSPLIT built-in cluster analysis approach. Total spatial variance (TSV) and the variance between each trajectory component were calculated to determine the optimum number of clusters (Draxler et al., 2009)."

Draxler, R. R., Stunder, B., Rolph, G., and Taylor, A.: HYSPLIT4 user's guide, http://ready.arl.noaa.gov/HYSPLIT\_info.php, 2009.

3. The authors suggest that because the site is above the PBL  $O_3$  levels before dawn (e.g., 2:00–5:00 LT) are representative of the regional background  $O_3$ . This could be the case, however all that can be said is that the site is dynamically disconnected from the PBL, how can they be sure that it is representative of the background?

Response: we are sorry that the terminology "regional background  $O_3$ " may be confusing. Here we want to use the nighttime  $O_3$  levels to **reflect** the regional  $O_3$  without impact of local photochemical production, because Mt. Tai is above the PBL and no photochemistry occurs at night. As pointed out by the reviewer, the nighttime  $O_3$  levels at Mt. Tai may be affected by the residual PBL air (and long-range transport of processed air masses), so they are a bit higher than the 'ideal' regional background  $O_3$  levels. For clarity, we have changed in the revised manuscript "regional background  $O_3$ " to "regional baseline  $O_3$ ", which is defined as the regional  $O_3$  without impact of local photochemical formation. Please see below for the revised definition.

"As the summit of Mt. Tai is well above the planetary boundary layer at nighttime, the  $O_3$  concentrations during the latter part of the night (e.g., 2:00–5:00 LT) are usually considered to reflect the regional baseline  $O_3$  (defined hereafter as regional  $O_3$  without impact of local photochemical formation)."

4. There is work by Clapp and Jenkin Atmospheric Environment 35 (2001) 6391–6405 that has shown that it is possible to quantify the regional contribution to oxidant (i.e. NOx independent) from plots of Ox vs. NOx. This work also looked at the contribution as a function of season too. This analysis should also be carried out for data presented in this present study and compare to the data presented in figure 7.

Response: we have conducted the same analysis to *Clapp and Jenkin* (2001) with our data, and also found increasing trend for the regional  $O_3$  that was derived from the scatter plots of  $O_X$  vs.  $NO_X$  (see the plot below). This result is consistent with our results as shown in Figure 7. As we only have a few  $NO_X$  measurements at Mt. Tai after 2006, it is somewhat hard to establish a statistical significant trend analysis with this method. So we don't show this analysis in the present study.

**Figure R1.** Regional background  $O_3$  mixing ratios at Mt. Tai in June and July-August over 2006-2015. The regional background  $O_3$  was determined from the  $O_X$  vs. NOX plot according to *Clapp and Jenkin* (2001).

Clapp, L. J., and Jenkin, M. E. Analysis of the relationship between ambient levels of  $O_3$ ,  $NO_2$  and NO as a function of NOx in the UK. Atmos. Environ., 35, 6391-6405, 2001.

5. The authors use HCHO as a marker for anthropogenic VOCs. There are not only anthropogenic sources of HCHO, there are also secondary in situ production routes, such as from CH4. Given that there are not only anthropogenic sources of HCHO, who will this impact on their analysis?

Response: we agree with the reviewer that there are not only primary sources but also secondary sources of HCHO. HCHO is a major oxidation product of a variety of VOC species, and secondary formation indeed presents a significant contribution to the total HCHO burden in the atmosphere. Therefore, increasing HCHO levels should imply the increase of VOC abundances (as HCHO precursors). For clarity, the following statement has been provided in the revised manuscript.

"Considering that HCHO is a major oxidation product of a variety of VOC species and due to the availability of the satellite-retrieved products, HCHO was selected as an indicator of the VOC abundances."

6. They are using satellite retrievals to infer PBL HCHO. Satellite retrievals in the PBL are notoriously difficult. What a priori assumptions have been made for their retrievals? More importantly what are the errors and what impact will it have on the results?

Response: we realize the large uncertainty of the HCHO satellite retrievals. In the present study, we primarily focus on the "trend", rather than the "absolute value" of HCHO column. We should also note that all data we have are only these satellite HCHO products to infer the regional VOC trends, given lack of in-situ VOC measurements. Moreover, the results derived from the satellite measurements agree well with those determined from the bottom-up VOC emission inventories, and are also consistent with the fact that the nationwide VOC controls have not been enforced so far in China. Despite the large uncertainty of satellite data, the major conclusion that the VOC emissions have been increasing in China should still be sound. We have added the following discussion in the revised manuscript.

"This result agrees very well with the emission inventory estimates which showed significant increases of anthropogenic VOC emissions in China in the past decades (Bo et al., 2008; Wang et

al., 2014), and is consistent with the lack of nationwide VOC controls. All of these results evidence the increase of atmospheric VOC abundances over the North China Plain."